# *Salmonella* Regulator STM0347 Mediates Flagellar Phase Variation via Hin Invertase

**DOI:** 10.3390/ijms23158481

**Published:** 2022-07-30

**Authors:** Hongou Wang, Zhiheng Tang, Baoshuai Xue, Qinghui Lu, Xiaoyun Liu, Qinghua Zou

**Affiliations:** 1Department of Microbiology, School of Basic Medical Sciences, Peking University Health Science Center, Beijing 100191, China; hongou.wang@bjmu.edu.cn (H.W.); zhihengt@pku.edu.cn (Z.T.); luqinghui@pku.edu.cn (Q.L.); 2Key Laboratory of Zoonosis, Ministry of Education, College of Veterinary Medicine, Jilin University, Changchun 130012, China; xuebs21@mails.jlu.edu.cn

**Keywords:** *Salmonella enterica*, STM0347, flagellar phase variation, Hin, motility, virulence

## Abstract

*Salmonella enterica* is one of the most important food-borne pathogens, whose motility and virulence are highly related to flagella. Flagella alternatively express two kinds of surface antigen flagellin, FliC and FljB, in a phenomenon known as flagellar phase variation. The molecular mechanisms by which the switching orientation of the Hin-composed DNA segment mediates the expression of the *fljBA* promoter have been thoroughly illustrated. However, the precise regulators that control DNA strand exchange are barely understood. In this study, we found that a putative response regulator, STM0347, contributed to the phase variation of flagellin in *S.* Typhimurium. With quantitative proteomics and secretome profiling, a lack of STM0347 was confirmed to induce the transformation of flagellin from FliC to FljB. Real-time PCR and in vitro incubation of SMT0347 with the *hin* DNA segment suggested that STM0347 disturbed Hin-catalyzed DNA reversion via *hin* degradation, and the overexpression of Hin was sufficient to elicit flagellin variation. Subsequently, the Δ*stm0347* strain was outcompeted by its parental strain in HeLa cell invasion. Collectively, our results reveal the crucial role of STM0347 in *Salmonella* virulence and flagellar phase variation and highlight the complexity of the regulatory network of Hin-modulated flagellum phase variation in *Salmonella*.

## 1. Introduction

As one of the most important food-borne pathogens, *Salmonella enterica* can cause vomiting, diarrhea, gastroenteritis, typhoid fever, systemic infection, and even death [1,2,3]. It is estimated that there are over 93.8 million cases of gastroenteritis caused by *Salmonella* infection per year worldwide, leading to 155,000 deaths from gastroenteritis and 84,800 deaths due to diarrhea every year [4,5]. In the process of infecting host cells, *Salmonella* first invades non-phagocytic intestinal epithelial cells to survive and proliferate intracellularly and then penetrates epithelial cells for deeper infection [6,7]. It is well documented that bacterial motility, chemotaxis, and virulence, including host cell invasion, depend on flagella to a large extent [8,9]. On the one hand, *Salmonella* flagella can drive bacteria to scan and colonize host cells [10,11]; on the other hand, they can also be recognized by the host innate immune system, triggering immune defenses [12]. Consequently, *Salmonella* has evolved specific mechanisms to evade host immunity.

*Salmonella enterica serovar* Typhimurium assembles a sophisticated flagellum structure. There are three basic components to the flagellum, including the basal body, hook, and filament, which alternatively expresses two different antigenic proteins, FliC (H1 antigen) and FljB (H2 antigen) [13,14]. The flagellin type exhibits a regulatory effect on the virulence of *Salmonella*. *Salmonella* locked in the FliC-expressing phase is more virulent than the FljB-expressing mutant, whether by oral or intraperitoneal inoculation, probably due to its superiority at infecting and colonizing the spleen [15]. Additionally, in normal swimming motility observations, there are no marked differences between FliC- and FljB-locked bacteria; however, FljB-expressing bacteria have a greater swimming speed under high-viscosity conditions [16]. An in-depth study revealed a distinct motor pattern between *Salmonella* with two flagella phases. When flagella were locked in the FliC-expressing phase, *Salmonella* tended to stop frequently and swim for a prolonged period near the host cell surface, which resulted in excessive gastrointestinal colonization in murine infection models [17].

The alternate expression of these two antigen phases of the flagellum is defined as flagellar phase variation, the study of which dates back to 1922 [18,19]. The regulatory mechanisms that underlie flagellum phase variation have also been widely reported. The gene that promotes *fljB* in *S*. Typhimurium is sliced by a reversible DNA segment, which is sandwiched by two repetitive sequences, *hixL* and *hixR* [20]. When this DNA segment is in one orientation and the promoter approaches the *fljBA* operon, the FljB flagellum is expressed, and *fliC* is suppressed at the transcriptional level by the direct binding of FljA with the *fliC* promoter [20]. When Hin recombinase controls the flipping of the reversible segments to another orientation, neither *fljB* nor *fljA* is transcribed, and the FliC flagellum dominates in the bacteria. Previous research has illustrated the necessity of Fis and HU binding with a recombinational enhancer on the *hin* gene while assembling the invertasome during DNA inversion [21,22]. However, studies that explore the other regulatory factors of Hin-mediated flagellum phase variation remain insufficient.

A previous study globally compared gene expression across *Escherichia coli*, *Salmonella enterica*, and *Bacillus subtilis*, and a phylogenetic tree analysis revealed a conserved homologous regulator of CsgD (*E. coli*), STM0347 (*S. enterica*) [23], which is a putative response regulator whose functions remain to be characterized. In the present study, we found that FliC was downregulated in the *stm0347* deletion mutant compared with the wild-type (WT) strain of *S*. Typhimurium via proteomic profiling. We further confirmed the flagellum phase variation phenomenon in the *stm0347* deletion mutant by visualizing secreted proteins. Notably, our data indicated STM0347-dependent Hin-inversion, suggesting a linkage between this transcription factor and *Salmonella* flagellum phase variation. Importantly, we provide a series of lines of evidence that STM0347 might prevent invertible DNA from flipping orientation by diminishing *hin* mRNA levels or degrading *hin* DNA fragments, and the overexpression of Hin was able to induce flagellar phase variation to a certain extent. In addition, a variant lacking *stm0347* was outcompeted by WT in terms of host cell invasion.

## 2. Results

### 2.1. Homology Analysis of STM0347 Protein

*Salmonella* STM0347 is a putative response regulator whose functions remain to be characterized. A genome comparison between *E. coli* and *S.* Typhimurium found homology between STM0347 and a LuxR-like protein, CsgD, and proposed that STM0347 might have evolved to be a regulator of pathogenesis-related protein [23]. STM0347 is relatively distinctive among bacterial species, and sequence alignment with its potential homologous protein also indicates that it exerts similarity to a transcriptional regulatory protein, UhpA, which is known to contain a helix-turn-helix (HTH) luxR-type domain (Figure 1A). UhpA belongs to the NarL family, exhibits a regulatory effect on sulfur assimilation pathways in *Salmonella*, and participates in sugar phosphate transport by directly binding to the *uhpT* promoter in *E. coli* [24,25]. In addition, STM0347 is highly conserved among numerous *Salmonella* strains according to a BLAST search against reference sequences in the NCBI database (some representative STM0347 homologues are shown in Figure 1B). Although the sequence analysis implied a C-terminal winged helix-like DNA-binding domain, the underlying regulon and its potential functions need to be further explored.

### 2.2. Comparative Proteomic Profiling of Wild-Type S. Typhimurium SL1344 and Its Isogenic Strain ΔSTM0347

To elucidate the STM0347 regulon in *S.* Typhimurium, we performed quantitative proteomic profiling of the *stm0347* deletion mutant compared with its parental WT strain. In total, 1802 proteins were identified from three biological replicates. A complete list of all identified proteins is provided in Appendix A. To globally depict differentially expressed proteins in these two strains, we developed a protein-level volcano plot complying with the criteria described in the “Materials and Methods” section (Figure 2). With the label-free quantitation (LFQ) strategy, 41 proteins in *stm0347* deletion mutants were differentially regulated, including 31 downregulated and 10 upregulated proteins, compared with WT (see Appendix A). The putative outer membrane lipoprotein STM0349 showed an outstanding lead in the upregulation of all proteins tested; however, its precise function still needs to be defined. We then noticed that the H1 antigen flagellin (FliC) and the flagella synthesis protein (FlgN) were significantly downregulated when *stm0347* was lacking. Interestingly, we also found the suppression of HilA, a well-known transcriptional regulator of the *Salmonella* pathogenicity island-1 (SPI-1) type III secretion system (T3SS) in *Salmonella*. These findings strongly suggest that STM0347 might play a regulatory role in mediating *Salmonella* virulence.

### 2.3. The Deprivation of STM0347 Leads to Flagellum Phase Variation in S. Typhimurium

To test the hypothesis that *stm0347* deletion induced the attenuation of *Salmonella* motility and virulence, we enriched the secreted proteins in WT and the Δ*stm0347* mutant and visualized the secreted pattern between these two strains by both Coomassie blue staining and immunoblotting. Interestingly, we observed a marked increase in the secretion of an approximately 52 kDa protein, as well as a reduction in one near 48 kDa in the *stm0347* deletion mutant (Figure 3A). We further authenticated the two bands excised from gel by LC–MS/MS and identified them as flagellin subunits FljB and FliC (see Appendix A). Additionally, we double checked the secretion of these two flagellin subunits with immunoblotting (Figure 3B). To confirm that the specification of the phenotype is triggered by *stm0347* deletion rather than genetic manipulation or other inscrutable factors, we constructed several deletion mutants, including two virulence factors (SptP and SseL) and two metabolic regulatory factors (EntC and AraA), and detected the flagella phases. The alternative expression of two flagella phases was not observed in these strains (Figure 3C). The compensation of STM0347 by exogenous expression plasmid or genome recombination of the *stm0347* coding sequence displayed a consistent flagella phase, FliC, with WT *Salmonella* (Figure 3A), which provided further evidence of the regulatory effect of STM3047 on the flagella phase variation.

### 2.4. ΔSTM0347-Induced Flagella Phase Variation Depends on the Modulation of Invertible Promoter of fljB

We subsequently explored whether the expression levels of *fljB* and *fliC* were altered in the *stm0347* deletion mutant at the transcriptional level. As is shown in Figure 4A, the deprivation of *stm0347* dramatically reduced the mRNA level of *fliC* and accompanied a surge in the *fljB* level. Correspondingly, complementation with the STM0347-expressing plasmid rescued the *fliC* and *fljB* transcripts back to WT levels (Figure 4A), suggesting a potential transcriptional regulatory effect of STM0347 on *fljB*. Given the putative role of the response regulator with a winged helix-like DNA-binding domain (see Figure 1), we next sought to test whether STM0347 exhibits a direct transcriptional regulatory role in *fljB* via an electrophoretic mobility shift assay (EMSA). DNA fragments containing the *fljB* promoter were incubated with purified 3×FLAG-tagged STM0347 protein; however, no retardation in electrophoretic mobility was shown at varying concentrations of STM0347 (Figure 4B).

We next focused our attention on other possible mechanisms. Since the Hin-involved DNA segment inversion participates in the flip-orientation of the *fljBA* promoter, and ultimately controls the alternative expression of *fliC* and *fljB* [20], the proportion of the two different orientations of the invertible segment were measured with real-time PCR and PCR amplification in the WT strain, in the *stm0347* deletion mutant, and under complement conditions. The primers were designed according to the principles of previous studies [27,28]. For real-time PCR, one primer was located outside the invertible region, and the other two primers were located within the invertible region, which made the PCR products of *hin*-on-orientation (FljB-expressed) and *hin*-off-orientation (FliC-expressed) both nearly 500 bp. Compared with the WT strain, the *hin*-on- to *hin*-off-orientation ratio was significantly elevated in the *stm0347* deletion mutant, while the ratio was restored by the heterogeneous expression of the STM0347 protein on the plasmid (Figure 5A). For the PCR amplification strategy, two recombination sites, *hixL* and *hixR*, were produced from two pairs of primers. In the Δ*stm0347* strain, the *hin*-off-orientation fragment could barely amplify, while the products of the *hin*-on-orientation fragment were markedly enhanced (Figure 5B). In addition, chromosomally complemented *stm0347* was adequate to resume the flip sequence switching back to WT levels. Interestingly, flagella phase variation did not occur when we locked the invertible sequence to one orientation or knocked out the coding sequence of *hin* at a specific orientation (Figure 5C,D). Together, these findings reveal that STM0347-triggered flagella phase variation is related to the switch orientation of the Hin-containing DNA segment.

### 2.5. ΔSTM0347-Induced Flagella Phase Variation Is Probably Involved in the Degradation of Hin

It has been confirmed that Fis and HU binding to Hin is an essential step to assemble the ‘invertasome’ during DNA strand exchange [29]. Thus, we first suspected that the STM0347 might suppress the expression of *fis*, *hin*, or the two subunits of HU, *hupA*, and *hupB*. We did not find any difference in the transcriptional levels of *fis*, *hupA*, or *hupB* among the WT, the *stm0347* deletion mutant, or Δ*stm0347* complemented with an STM0347-expressing plasmid (Figure 6A). Notably, *stm0347* deletion led to a statistically significant upregulation of the mRNA levels of *hin*. Due to the low endogenous expression level of STM0347, knockout of *stm0347* did not induce a dramatic change in *hin*. In addition, when we tried to induce the exogenous STM0347-expressing plasmid close to the endogenous expression level with 0.001% (*wt*/*vol*) arabinose, *hin* expression was attenuated to the WT level (Figure 6B). On the other hand, we hypothesized that STM0347 might disturb Fis-initiated invertasome assembly via competitive binding to a Hin enhancer; thus, an EMSA was conducted to test whether STM0347 directly binds to the *hin* DNA segment. Interestingly, we observed a decrease in the free DNA band without a corresponding shift in band binding with STM0347 proteins (Figure 6C). Combined with the qPCR results, we suspected that sufficient amounts of STM0347 protein could degrade the *hin* gene cassette, and consequently restrain the DNA inversion. To explore this possibility, we further introduced excessive Hin proteins by exogenous plasmids (0.2% arabinose) into the WT strain, and we inspected whether flagella phase variation occurred. As shown in Figure 6D, overexpression of Hin could elicit a portion of flagella transformation from FliC to FljB. Taken together, these data indicated that STM0347 might inhibit DNA inversion by degrading *hin* DNA segments, ultimately preventing flagella phase variation to a certain extent.

### 2.6. ΔSTM0347 Deletion Disadvantages S. Typhimurium for Cell Invasion

Based on our observation that STM0347 regulated flagellin expression at both the transcriptional and translational levels, we sought to investigate whether STM0347 exerts a regulatory role in *Salmonella* motility. Swimming and swarming motility were assessed with 0.3% and 0.7% soft-agar plates. Not unexpectedly, there were barely any differences in terms of motility between the *stm0347* deletion mutant and its parental WT strain (Figure 7A,B), probably because STM0347 mainly mediates flagellin alteration rather than suppressing overall flagellar expression. The differences in the structure and dynamics of these two flagellar filaments might impact *Salmonella* motility under specific circumstances [16]. It was shown that FljB-expressed *Salmonella* (Δ*stm0347*) had a slightly greater swimming motility under high viscosity conditions compared with FliC-expressed bacteria (WT) (Appendix A). Given that FliC and FljB represent two different antigen flagellins [14], flagellar phase variation might be involved in immune evasion during host infection. We exploited a HeLa-cell competitive invasion assay using a 1:1 ratio mixture of WT *Salmonella* and the *stm0347* deletion mutant, and colony forming units (CFUs) were calculated 2 h post-infection. Meanwhile, to rule out the impact of growth rate differences on invasion efficiency, we monitored the growth situation of those two strains over a 10 h period, and Δ*stm0347* appeared to be indistinguishable from WT *Salmonella* (Figure 7C). The results indicated that the WT strain remarkably outcompeted the *stm0347* deletion mutant (Figure 7C). Together, these data suggest that the absence of *stm0347* does not affect *S.* Typhimurium motility; however, it is disadvantageous for host cell invasion.

## 3. Discussion

The goal of our current study was to define the functions and regulatory mechanisms of STM0347 in *Salmonella* virulence. Our proteomics data revealed remarkable inhibition levels of HilA and FliC in the *stm0347* deletion mutant compared with the parental WT strain. The pathogenesis of *Salmonella* mainly depends on effector proteins, which are encoded by *Salmonella* pathogenicity islands (SPIs) on its genome, and transported into the host cell cytoplasm by the T3SS [30]. HilA has been widely reported to have a direct activation effect on two promoters of SPI-1, which plays a critical role in *Salmonella* invasion during the early stage of infection [31]. In a further investigation of the *stm0347* deletion mutant secretion protein patter, we observed that *stm0347* absence induced flagellin expression in *S.* Typhimurium to alter from FliC to FljB, which relied on the invertible expression of the *fljB* promoter flanked by a reversible DNA segment.

There are two flagellins, FliC and FljB, expressed in *Salmonella*, and they switch at a frequency rate of 10^−3^ to 10^−5^ per cell per generation [32]. Only FljB flagellin is produced upon *fljBA* operon transcription, and it further inhibits *fliC* expression by directly binding to its promoter region; meanwhile, only FliC flagellin is present when the operon inverts to the other orientation [33]. In this study, *stm0347* deprivation drastically decreased the expression of FliC and increased the expression of FljB at the transcriptional and translational levels. Meanwhile, our data showed that Δ*stm0347* induced flagellar phase variation via a Hin-participating DNA segment flipping reaction. Interestingly, in addition to STM0347, IacP and QseG have also been reported to regulate flagellar phase variation by Hin inversion in *Salmonella* [34,35]; however, an in-depth understanding of the underlying regulatory mechanisms requires further investigation.

Three critical steps are essential for Hin-catalyzed DNA exchange [29,36,37]. First, the Fis-bound enhancer initiates the assembly of the invertasome, which is also known as a γδ resolvase topological structure. Afterwards, the Hin dimer binds to the synapsed *hix* site invertase subunits and covalently joins each 5′ end of the broken strands via an ester linkage at serine 10. Ultimately, DNA strand exchange occurs under catalytic conditions. During invertasome assembly, the presence of either of two HU subunits, HupA or HupB, is required to bind with a nonspecific sequence between the enhancer and recombination site [22]. In this case, the mRNA levels of *fis*, *hupA*, and *hupB* were unchanged after *stm0347* deletion. Remarkably, the *hin* transcription level was significantly increased after *stm0347* knockout (approximately twofold) and was attenuated by complementing the exogenous STM0347 protein. The growth of BL21 was inhibited when GST-tagged STM0347 was expressed on pGEX-6p-1 plasmid, and inclusion body proteins formed when 6×His-tagged STM0347 was expressed on pET-28a plasmid (data not shown). Thus, we suspected that STM0347 could only be expressed at a relatively low level in *Salmonella*, so endogenous knockout of *stm0347* simply caused a limited fold change in the *hin* expression level. However, despite the modest regulatory levels, *stm0347* deletion was still adequate for the alteration of flagellin expression with a biological amplification cascade. Combined with our in vitro incubation results, flagellar phase variation was probably arrested by the degradation of Hin when STM0347 was present in WT *Salmonella*. We further discovered that overexpression can also induce the alternative expression from FliC to FljB.

Although with highly homologous sequences and identical helical parameters, the distinguished structure and dynamics of the D3 domains mediates different motility functions in FliC and FljB flagellin-expressing *Salmonella* [16]. Similar to previous studies, no difference in motility was detected between the *stm0347* deletion mutant and the WT strain [17,34,35]. Since we only observed the phase variation of flagella, rather than the expression of total amount of flagella, this phenotype was not entirely unexpected. Furthermore, flagellin is a major surface antigen for many bacterial species, and both purified recombinant FliC and FljB protein can comparably activate the NF-κB pathway and induce interleukin 1β (IL-1β) secretion from host cells [38,39]. In this study, we found that the invasion of the *stm0347* deletion mutant was significantly outcompeted by the WT stain during the competitive invasion of HeLa cells. This was most likely because the WT strain, which expressed mainly FliC flagellin, had a distinct advantage in scanning the target cell surfaces and near-surface motility behavior as an immune evasion strategy [17]. HilA has been defined as a master regulator for the SPI-1 gene cluster, which plays critical roles in *Salmonella* invasion and virulence [40,41,42]. A previous study also showed that knockout of *hilA* in *S.* Enteritidis resulted in the repression of the cell invasion ability [43]. On the other hand, in clinically isolated antibiotic resistant *Salmonella* strains, *invA* (100%) was the most spread gene, followed by *hilA* (88.24%), *stn* (58.82%), and *fliC* (52.94%) [44]. Consistent with these theories, our data showed that the decrease in HilA protein levels was at least partially responsible for the impairment of invasion ability in the Δ*stm0347* strain.

In conclusion, the present work took advantage of high-resolution mass spectrometry to quantitatively profile the protein expression of a *Salmonella* mutant lacking *stm0347* compared to its parental strain, which depicted the biological functions of STM0347. Notably, we proposed that STM0347 diminished the Hin level, which inhibited the inversion of the Hin-mediated DNA segment, and further prevented flagellin expression from alternating from FliC to FljB (Figure 8). Importantly, the loss of STM0347 might induce, or at least partially contribute to, *Salmonella* virulence attenuation during host cell invasion. Our study compensates for the lack of research on the biological functions of STM0347 and highlights the complexity of STM0347′s regulation of Hin-mediated flagellum phase variation in *Salmonella*.

## 4. Methods and Materials

### 4.1. Bacterial Strains, Mutant Construction and Molecular Cloning

The *S.* Typhimurium strain SL1344 was used in this study, and all other strains are detailed in Appendix A. The bacteria were routinely cultivated at 37 °C in Luria–Bertani (LB) plates with 2% agar and 30 μg/mL streptomycin, unless otherwise specified. The *stm0347* deletion mutant (Δ*stm0347*) was constructed by the λ red recombinase system based on the protocol of a previous report [45]. Briefly, a kanamycin resistance gene cassette with 56 bp homologous fragments of *stm0347* at each end was amplified from pKD4 plasmids and electroporated into SL1344 carrying pKD46. The resulting *stm0347* deletion mutant with kanamycin resistance was preserved for the following competitive infection assay. To further generate chromosomally compensated *stm0347*, the phase lock mutants, and *hin* deletion mutants with the λ red system, the kanamycin resistance gene cassette in Δ*stm0347* was eliminated with the pCP20 plasmid. Successfully deleted target genes or mutated DNA segments were verified by both PCR and sequencing analysis. To construct an STM0347 complementation plasmid, the *stm0347* fragment was amplified, restriction digested, and ligated into the pSB3313 plasmid with an arabinose-inducible promoter and a C-terminal 3×FLAG tag. All primers used in this study are listed in Appendix A.

### 4.2. Proteomic Sample Preparation and LC–MS/MS Detection

To explore the underlying STM0347-regulated targets, we conducted proteomic analyses of Δ*stm0347* along with the WT strain. In brief, a single colony of each strain was inoculated in LB broth with 30 μg/mL streptomycin, and then the overnight culture was passaged 1:20 into fresh LB broth and harvested when the optical density of 600 nm reached 0.9. After centrifugation at 8000× *g* for 5 min and washing with ice-cold PBS 3 times, the resulting bacterial pellets were resuspended in SDS–PAGE loading buffer and denatured at 100 °C for 5 min. Protein samples were prefractionated using 10% SDS–PAGE and separated into 6 fractions, and in-gel trypsin digestion was performed according to a previous description [46].

The resulting peptides were dissolved in HPLC-grade water prior to LC–MS/MS analyses. For label-free proteomics analyses, a nanoflow reversed-phase LC (nLC, Thermo Fisher Scientific, Waltham, MA, USA) coupled with a hybrid ion trap orbitrap mass spectrometer (LTQ Orbitrap Velos, Thermo Scientific, Waltham, MA, USA) was applied for peptide analyses. Homemade C18 analytical columns (75 μm × 150 mm) were equipped with 4 μm of 100 Å Magic C18AQ silica-based particles (Michrom BioResources Inc., Auburn, CA, USA). A 47-min gradient was employed at the start with 5% solvent B (100% ACN, 0.1% FA), which gradually rose to 45% within 40 min. Then, the concentration of solvent B ascended to 95% in 5 min, which was followed by elution with 95% solvent B for the last 2 min. Eluted peptides were electrosprayed directly into the mass spectrometer for MS and MS/MS analyses in data-dependent acquisition mode. The full MS scan was (*m*/*z* 350–1200) exploited, and the 10 most intense ions were chosen for subsequent MS/MS analyses.

### 4.3. Proteomic Data Processing

The raw files were processed with MaxQuant (http://maxquant.org/, version 1.5.3.8, accessed on 18 November 2021) based on the combination *S.* Typhimurium protein database of LT2 (Taxon identifier: 99287) and SL1344 (Taxon identifier: 216597) downloaded from the UniProt website (www.uniprot.org, accessed on 25 June 2020) [47]. The precursor mass tolerance and the fragment mass tolerance were set at 20 ppm and 0.8 Da, respectively. Meanwhile, methionine oxidation was set as a variable modification. Trypsin was set as a digestion enzyme with a maximum of two missed cleavages. The false discovery rates (FDRs) of peptides and proteins were restricted to under 1%. We then filtered the proteins that were only identified by site, matched the reverse database, and contained potential contaminants with Perseus, and we also replaced the missing values with random numbers [48]. Two-group comparisons were performed with paired Student’s *t*-tests. Proteins with *p* < 0.05 and average fold change > 2.0 were considered significantly regulated.

### 4.4. Western Blot for Secreted Proteins

The supernatants of WT and Δ*stm0347* were collected after centrifugation at 7000× *g* for 20 min and filtered with a 0.22 μm membrane to further remove the remaining bacteria. Sodium deoxycholate was added to a final concentration of 0.2% (*wt*/*vol*) before the secreted proteins were precipitated with 20% (*wt*/*vol*) trichloroacetic acid (TCA) overnight at 4 °C. Finally, the protein pellets were collected, washed with ice-cold acetone 3 times, and denatured in SDS–PAGE loading buffer with 100 mM Tris-HCl (pH 8.0) at 95 °C for 10 min. Secreted protein samples were loaded for SDS–PAGE electrophoresis and were either transferred onto polyvinylidene difluoride membranes (Millipore, Billerica, MA, USA) or stained with Coomassie blue. For immunoblotting, membranes were incubated with antibodies and visualized with ECL Western Blotting Substrate (Tanon, Shanghai, China). Antibodies and dilutions were as follows: FLAG (Zen-bio, Chengdu, China, #T201126-3A6, 1:5000) and FliC (Abcam, Cambridge, UK, #ab93713, 1:5000).

### 4.5. RNA Extraction and Quantitative Real-Time PCR

Bacterial cells, including WT, *stm0347* deletion mutant, and Δ*stm0347* complemented with an STM0347-expressing plasmid, were collected at OD_600_ = 0.9 and ground in liquid nitrogen. Total RNA was extracted with TRIzol™ Reagent (Invitrogen™, Grand Island, NY, USA) and reverse transcribed to cDNA using the PrimeScript™ RT–PCR Kit (Takara, Dalian, China). Real-time PCR was performed on a 7500 Fast Real

-Time PCR system (Applied Biosystems™, Foster City, CA, USA), which was collocated with a SYBR Premix EX Taq II Kit (Takara, Dalian, China) to quantify *fljB*, *fliC*, *fis*, *hupA*, *hupB*, and *hin* at the transcriptional level. We used 16S ribosomal RNA as a housekeeping gene. The primers used for real-time PCR are listed in Appendix A.

### 4.6. DNA Extraction and Hin-Mediated Invertible Segment Orientation Determination

To explore the induction of flagellar phase variation, we picked 4 primers based on a previous depiction [28] to determine the orientation of the Hin-mediated invertible segment and the *fljBA* promoter region. WT, Δ*stm0347*, and Δ*stm0347*::*stm0347* bacteria were harvested at OD_600_ = 0.9, and total DNA was extracted using an Ezup Column Bacteria Genomic DNA Purification Kit (Sangon, Shanghai, China) according to the manufacturer’s instructions. DNA concentrations were measured with a NanoDrop™ spectrophotometer (Thermo Scientific, Waltham, MA, USA). Equal amounts of DNA were loaded for PCR amplification, and the primers and PCR product sizes were as follows: *hixL* of *hin*-off-orientation (*hix-1*/*hix-2*, 550 bp), *hixR* of *hin*-off-orientation (*hix-3*/*hix-4*, 1700 bp), *hixL* of *hin*-on-orientation (*hix-1*/*hix-3*, 1137 bp), and *hixR* of *hin*-on-orientation (*hix-2*/*hix-4*, 1113 bp). To quantify the proportion of the on or off orientation of the invertible segment in each bacterial strain, real-time PCR was performed as described above.

### 4.7. Recombinant Protein Expression and Purification

FLAG-tagged STM0347 was expressed in the *S.* Typhimurium WT + *p*STM0347 strain. Briefly, 400 mL bacterial suspensions were cultured to OD_600_ = 0.6, and then 0.4% arabinose was added and induced for 3 h. Bacterial cells were lysed in 30 mL of ice-cold PBS buffer via sonication, and cell lysates were clarified by centrifugation at 7000× *g* for 20 min at 4 °C and filtered through a 0.45 μm membrane. The resulting supernatants were incubated with anti-FLAG M2 agarose beads (50% slurry, Sigma-Aldrich, St. Louis, MO, USA) for 4 h at 4 °C to capture FLAG-tagged STM0347. Afterwards, the beads were washed 3 times with a washing buffer containing 25 mM Tris-HCl (pH 7.5) and 150 mM NaCl. FLAG peptides were used to elute the STM0347 protein and were removed by ultrafiltration.

### 4.8. Electrophoretic Mobility Shift Assays (EMSAs)

DNA fragments of the putative promoter region of the *fljBA* operon and the coding sequence of *hin* were amplified by PCR, purified with a Gel Extraction Kit (Axygen^®^, Corning, CA, USA), and dissolved in water. Then, purified STM0347 proteins were incubated with 30 ng DNA fragments in 20 μL of binding buffer (10 mm Tris-HCl (pH 8.3), 50 mM KCl, 1 mM MgCl_2_, 0.5 mM DTT, 0.5 mM EDTA, 50 μg/mL BSA and 0.2% vol/vol glycerol). The contents of STM0347 were set at 0, 0.5, 2, and 5 μg. The reaction mixtures were incubated at room temperature for 20 min and then loaded onto 6% native polyacrylamide gels for electrophoresis in 1×TAE buffer (40 mM Tris, 20 mM acetic acid and 1 mM EDTA) in an ice bath. The gel was then stained with Goldview Nucleic Acid Gel Stain (Yeasen, Shanghai, China) or Coomassie blue and photographed using the ChemiDoc™ imaging system (Bio-Rad, Woodinville, WA, USA).

### 4.9. Competitive Invasion Assay

The HeLa cells were cultured in Dulbecco’s modified Eagle medium (DMEM, HyClone, Logan, UT, USA) supplemented with 10% fetal bovine (TransGen, Beijing, China) at 37 °C under 5% CO_2_ conditions. Cells were ready for the invasion assay when the density approximated 80–90% confluence. The overnight cultured WT *Salmonella* and *stm0347* deletion mutant with kanamycin resistance were diluted 1:20 until the OD_600_ reached 0.9, and were then mixed at a 1:1 ratio. Cells were preequilibrated with Hanks’ buffered salt solution (HBSS) and infected with a mixed bacterial suspension at a multiplicity of infection (MOI) of 10 for 1 h. Then, the cells were washed 3 times with HBSS and treated with 100 μg/mL gentamicin to kill extracellular bacteria. Finally, the cells were lysed and plated on streptomycin- or kanamycin-supplemented LB plates to calculate colony forming units (CFUs).

### 4.10. Growth and Motility Study

To obtain the growth curve of WT *Salmonella* and the *stm0347* deletion mutant, overnight cultured bacteria were diluted 1:50 in LB broth, and the optical density was monitored at 600 nm hourly over a 10 h period.

Swimming motility was detected as previously described [49] with slight modification. Briefly, bacteria were diluted 1:20 when the OD_600_ reached 0.9, and 2 μL was pipetted onto the middle of an LB plate with 0.3% agar. The plates were incubated at 37 °C, and the swimming zone was measured at 1, 3, and 8 h. To test the swimming motility under various viscosities, Ficoll PM400 (Sigma-Aldrich, St. Louis, MO, USA) was added to the swimming plates at a final concentration of 5% or 10% according to a previous description [16]. Swarming motility was analyzed on LB plates with 0.7% agar [50], which were cultured in closed containers with 100% humidity for 48 h.

### 4.11. Statistical Analysis

Data were collected from at least three independent tests, and the results are expressed as the mean ± SD. Statistical analyses were processed with SPSS 19.0 software (IBM, Armonk, NY, USA). Two-group comparisons were implemented with Student’s *t* test, while multiple-group comparisons were conducted with one-way ANOVA, which was followed by LSD *t* tests for between-group comparisons. Values of *p* < 0.05 were defined as statistically significant.

## Figures and Tables

**Figure 1 ijms-23-08481-f001:**
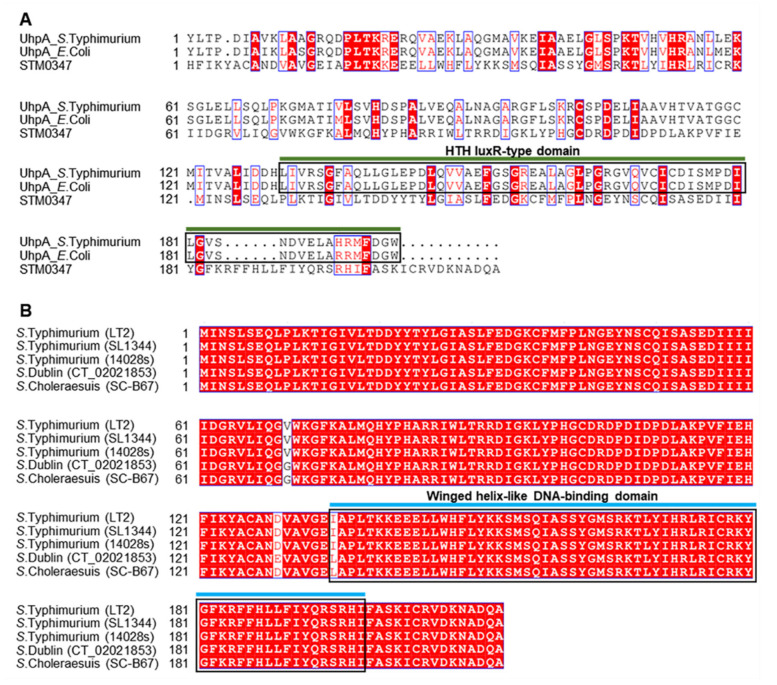
Homology analysis of STM0347 protein. Homologous proteins STM0347 were blasted with the NCBI database (https://blast.ncbi.nlm.nih.gov/Blast.cgi, accessed on 16 October 2021), multiple sequences of STM0347 with its homologs from different bacteria or different Salmonella strains were aligned by the CLUSTALW program (http://www.genome.jp/tools/clustalw/), and the figures were portrayed with ENDscript Web server (https://espript.ibcp.fr/ESPript/cgi-bin/ESPript.cgi, accessed on 16 October 2021) [26].

**Figure 2 ijms-23-08481-f002:**
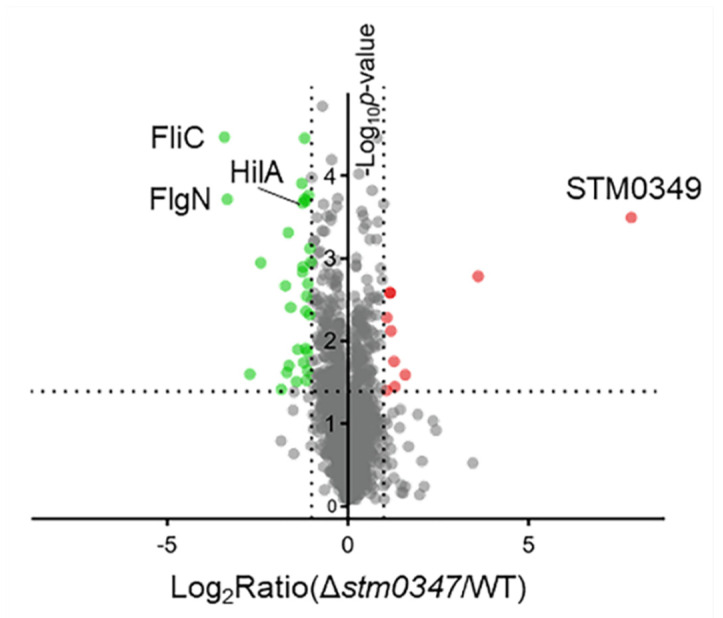
A volcano plot of *Salmonella* WT and Δ*stm0347* strains detected by LC–MS/MS analysis. The logarithmic values of the abundance ratios of Δ*stm0347* to WT was reported on the *x*-axis. The negative logarithmic *p*-values processed with *t* test was presented on the *y*-axis. Dotted lines denote two-fold (vertical) and *p* < 0.05 cutoff (horizontal). Data was collected from three biological replicates.

**Figure 3 ijms-23-08481-f003:**
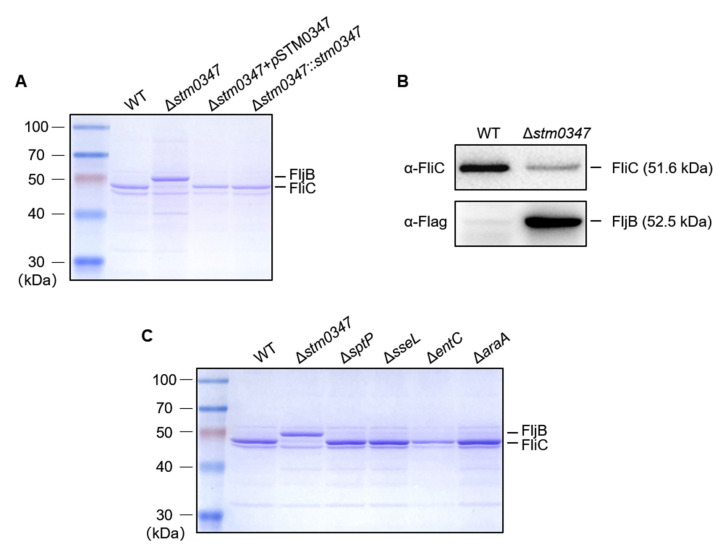
STM0347 regulates flagellin phase variation in *S.* Typhimurium. (**A**) Secreted proteins in the wild-type strain, the *stm0347* deletion mutant strain, the Δ*stm0347* complemented with an STM0347-expressing plasmid strain, and the Δ*stm0347* chromosomally complemented with *stm0347* coding sequence strain, were visualized with Coomassie blue strained gel. (**B**) Protein levels of FliC and FljB in secreted proteins of the WT and *stm0347* deletion mutant strain, determined with Western blot. (**C**) Secreted proteins in the WT strain, the *stm0347* deletion mutant strain, and the *sptP*, *sseL*, *entC*, and *araA* deletion mutant strain were visualized with Coomassie blue strained gel.

**Figure 4 ijms-23-08481-f004:**
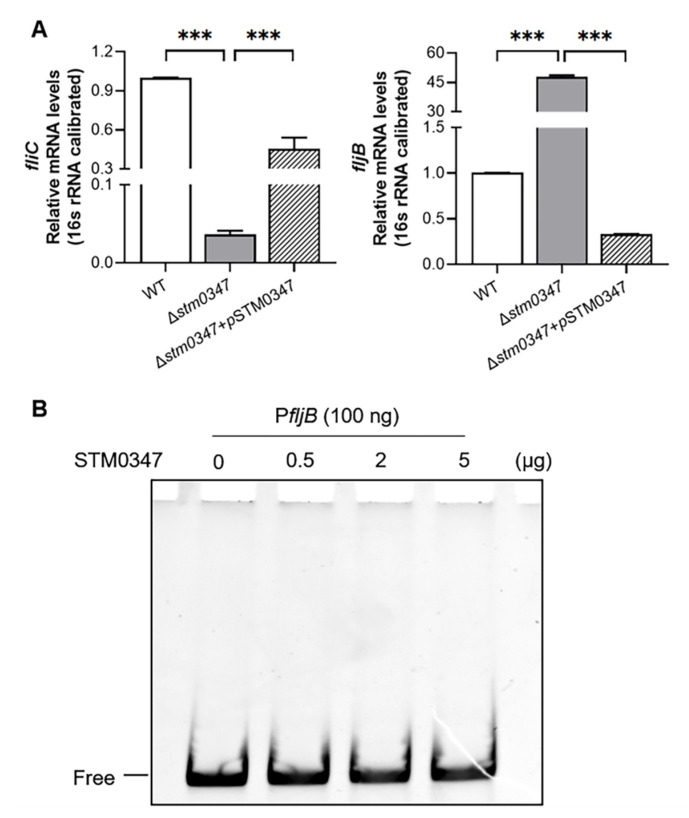
Transcriptional regulatory effect of *fliC* and *fljB* by STM0347. (**A**) Relative mRNA levels of *fliC* and *fljB* in WT, Δ*stm0347*, and Δ*stm0347* complemented with an STM0347-expressing plasmid induced with 0.001% arabinose. Expression levels are calibrated with 16 s rRNA. Results are presented as mean ± SD. Asterisks indicate significant differences (*** *p* < 0.001). (**B**) EMSA experiments denied the direct regulation of STM0347 on *fljB* promoter. Different amounts of purified STM0347 (ranging from 0 to 5 μg) were incubated with *fljB* promoter prior to electrophoretic separation. ‘Free’ indicates free DNA.

**Figure 5 ijms-23-08481-f005:**
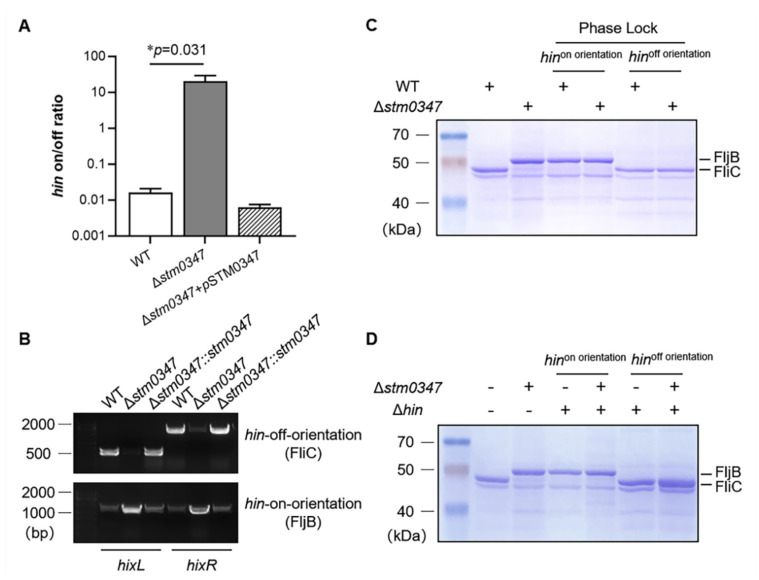
The alternation orientation of the *hin*-composed DNA segment was related to the STM0347-mediated flagella phase variation. (**A**) A real-time PCR strategy determined the ratio of *fliC*- and *fljB*-expressing DNA orientation in WT, Δ*stm0347* and Δ*stm0347* complemented with a STM0347-expressing plasmid induced with 0.001% arabinose. Results are presented as mean ± SD. Asterisks indicate significant differences (* *p* < 0.05). (**B**) Two *hix* sites representing the off orientation (top) or the on orientation (bottom) were PCR amplified from the genomic DNA of the WT, Δ*stm0347*, Δ*stm0347* chromosomally complemented with *stm0347* coding sequence strains, to identify the orientation of the *fliC*- and *fljB*- expressed DNA orientation. The sizes of PCR products were listed as follows: *hixL* of *hin*-off-orientation (550 bp), *hixR* of *hin*-off-orientation (1700 bp), *hixL* of *hin*-on-orientation (1137 bp), *hixR* of *hin*-on-orientation (1113 bp). (**C**) Secreted proteins in WT, the *stm0347* deletion mutant strain and four phase lock strains, was visualized with Coomassie blue strained gel. (**D**) Secreted proteins in WT, the *stm0347* deletion mutant strain and four *hin* deletion mutant strains, were visualized with Coomassie blue strained gel.

**Figure 6 ijms-23-08481-f006:**
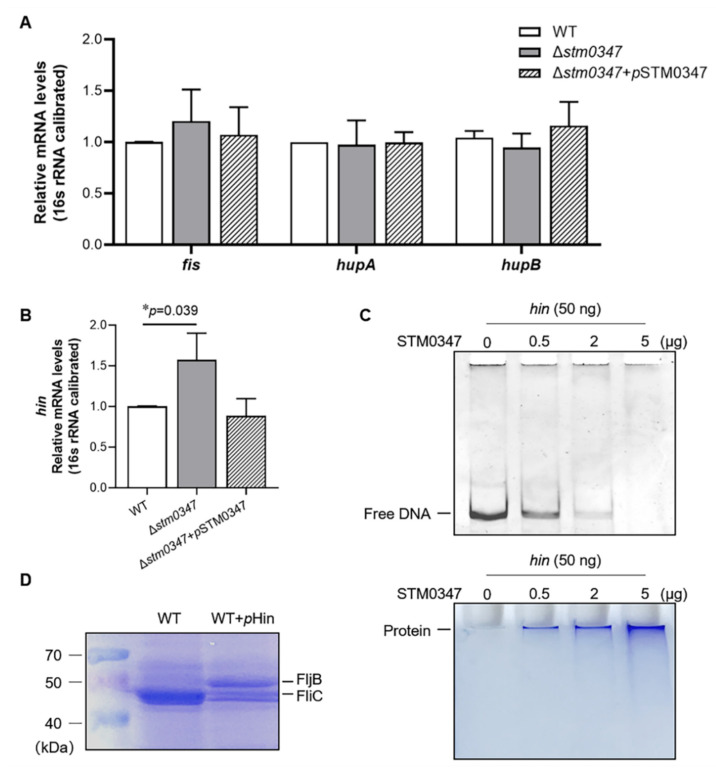
STM0347 inhibits Hin-catalyzed DNA reversion through diminishing *hin* level. (**A**) Relative mRNA levels of *fis*, *hupA*, and *hupB* in WT, Δ*stm0347*, and Δ*stm0347* complemented with a STM0347-expressing plasmid induced with 0.001% arabinose. Expression levels are calibrated with 16s rRNA. Results are presented as mean ± SD. (**B**) Relative mRNA levels of *hin* in WT, Δ*stm0347*, and Δ*stm0347* complemented with a STM0347-expressing plasmid induced with 0.001% arabinose. Expression levels are calibrated with 16s rRNA. Results are presented as mean ± SD. Asterisks indicate significant differences (* *p* < 0.05). (**C**) In vitro incubation of STM0347 and *hin* DNA segment. Different amounts of purified STM0347 (ranging from 0 to 5 μg) were incubated with *hin* DNA segment prior to electrophoretic separation. ‘Free’ indicates undegraded DNA bands. Gels were stained with Goldview (up) or Coomassie blue (down). (**D**) Secreted proteins in WT, and WT complemented with a Hin-expressing plasmid induced with 0.2% arabinose, were visualized with Coomassie blue.

**Figure 7 ijms-23-08481-f007:**
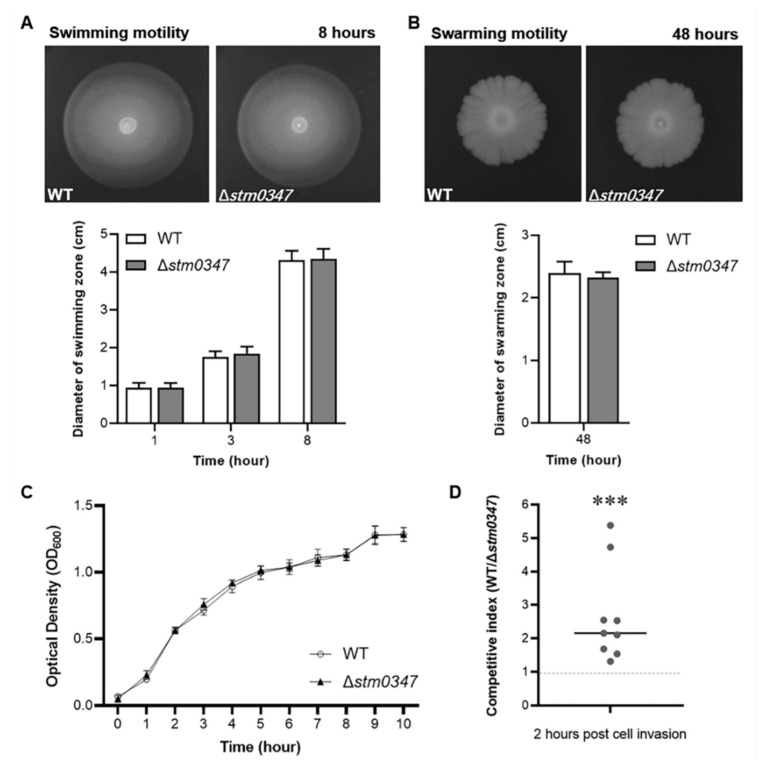
The absence of *stm0347* does not affect *S.* Typhimurium motility, however disadvantageous for host cell invasion. (**A**) Swimming motility of WT and Δ*stm0347* strains were investigated on the 0.3% agar LB plate, and the diameters of the swimming zones were monitored at 1, 3, and 8 h. Data were presented as mean ± SD. *n* = 12. (**B**) Swarming motility of WT and Δ*stm0347* strains were detected on the 0.7% agar LB plate, and the diameters of the swarming zones were measured at 48 h. Data were presented as mean ± SD. *n* = 12. (**C**) Growth curves of WT and *stm0347* deletion mutants in LB broth. Data were presented as mean ± SD. *n* = 6. (**D**) The competitive invasion between WT and Δ*stm0347* strains. HeLa cells were infected with a 1:1 mixture of streptomycin resistance WT strain and kanamycin resistance Δ*stm0347* strain at a multiplicity of infection (MOI) of 10, and CFU was determined 2 h post infection. (*n* = 9). Asterisks indicate significant differences (*** *p* < 0.001).

**Figure 8 ijms-23-08481-f008:**
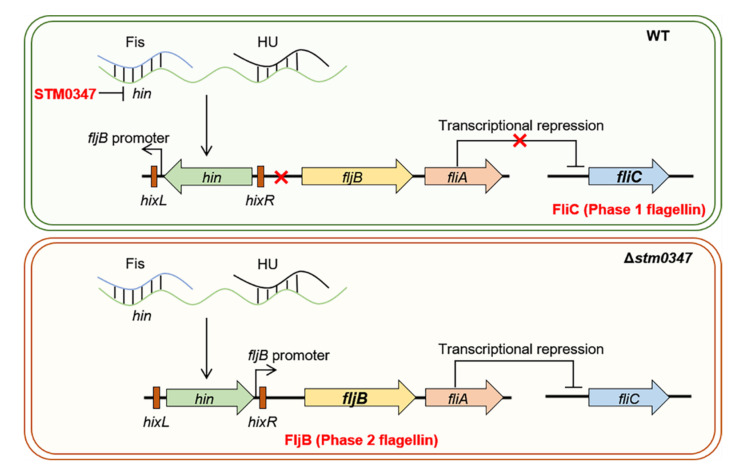
A proposed regulatory pathway of STM0347 mediated flagellin phase variation. STM0347 disrupted the Hin-catalyzed DNA strand exchange through attenuation of hin levels, and further inhibited the expression of flagellin transforming from FliC to FljB.

## Data Availability

The proteomics data presented in this paper have been deposited to the iProX database (URL: http://www.iprox.org/page/HMV006.html, accessed on 27 June 2022) under the accession number IPX0004628000.

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
