# Peer review of "Salmonella Regulator STM0347 Mediates Flagellar Phase Variation via Hin Invertase"

_ijms, 2022, doi:10.3390/ijms23158481_

Round 1
Reviewer 1 Report
In this paper, titled “Salmonella enterica Serovar Typhimurium putative response 2 regulator STM0347 mediates flagellar phase variation via Hin invertase”, Wang et al, aim to describe the role of STM0347. This work is interesting but needs careful proofreading as many mistakes can be found. For example, the authors are using FliB instead of FljB. It’s making the manuscript hard to follow as FliB is associated with Salmonella flagella too. Furthermore, more work on phenotypic characterization should be included. For these reasons, the paper needs to be thoroughly improved.
“serovar” does not need to be in italic and to have an uppercase.
Lines 19-20: “transforming” is vague.
Lines 40-41: rephrase as the flagella are not recognized by the immune system but only flagellins.
Line 46: “expressed on filament” should be rephrased, it’s constituting mainly the filament.
Line 49: rephrase
Line 60: it’s fljB and not fliB. Same remark on figure 3B, line 157.
Lines 138-139: rephrase
Line 199: rephrase
In figure 6, an experiment testing the motility of the WT and Dstm0347 under high viscosity should be added as described in ref 16.
Line 259: remain and not reminds
Lines 260-262: rephrase
Lines 262-263: STM0347 is distinct or similar to other bacterial genomes? Make it clear.
The first part of the discussion (until line 280) is still a result and should be moved.
Globally the discussion needs to be carefully proofread. Many words are inappropriate, and the English are very poor making it barely readable.
Line 269: Salmonella enterica strains
Line 285-287: rephrase
Line 343: spectrometry
Line 338: No italic for Enteritidis because it’s a serovar
Reviewer 2 Report
The manuscript by Wang and colleagues reports an extensive analysis about the role of putative STM0347, a homologous regulator of CsgD in E. coli, in the phase variation of flagellin in S. Typhimurium. The characterization of STM0347 protein is excellent both a transcriptional and translational level and its biological significance in Salmonella virulence could be relevant. However, more studies about the implication of this protein in cell invasion might be needed.
In my opinion, I recommend this study for publication.
Author Response
We greatly appreciate your comments on our manuscript (ijms-1816709). My co-authors and I are grateful to the referees for pointing out the shortcomings of the manuscript. In this study, we mostly focused our attention on the regulatory effects of STM0347 of flagella in Salmonella, and the motility mediated by which. We will take the implication of this protein in cell invasion into consideration in our following study, and a detailed research about the impacts of STM0347 on Salmonella invasion and infection will be conducted in future.
Round 2
Reviewer 1 Report
Before final acceptance, the authors should revise the Ficoll concentrations between the supp fig 2 and the Methods and Materials. 15% is not on the supp fig 2.
Author Response
Thanks very much for your comments.
We have revised the Ficoll concentrations in the "Methods and Materials" section: To test the swimming motility under various viscosities, Ficoll PM400 (Sigma-Aldrich, USA) was added to the swimming plates at a final concentration of 5% or 10% according to a previous description [16]. (Line 490-492)